# Age and Sex Comparisons in Pediatric Track and Field Hurdle Injuries Seen in Emergency Departments of the US

**DOI:** 10.3390/sports11030065

**Published:** 2023-03-10

**Authors:** Jacob Jones, Luke Radel, Kyle Garcia, David Soma, Shane Miller, Dai Sugimoto

**Affiliations:** 1Scottish Rite for Children, Dallas, TX 75219, USA; 2Departments of Orthopaedic Surgery and Pediatrics, University of Texas Southwestern Medical Center, Dallas, TX 75390, USA; 3Mayo Clinic, Rochester, MN 55905, USA; 4Department of Urology, University of Arizona College of Medicine, Tucson, AZ 85724, USA; 5Faculty of Sport Sciences, Waseda University, Tokyo 202-0021, Japan; 6The Micheli Center for Sports Injury Prevention, Waltham, MA 02453, USA

**Keywords:** hurdles, hurdle injuries, pediatrics, injury sex differences, injury age differences, track and field

## Abstract

There is limited literature analyzing pediatric hurdle injuries based on sex and age. This study compares hurdle-related injury types, injured body parts, and injury mechanisms by age and sex in pediatrics. Hurdle-related injury data from the National Electronic Injury Surveillance System were used to retrospectively review the injuries of hurdlers 18 years and under. Differences in injured body parts, injury types, and mechanisms were analyzed by age (pre-high school vs. high school) and sex (male vs. female). A total of 749 cases were extracted. Fractures were more common in pre-high schoolers (34.1% vs. 21.5%, *p* = 0.001), while more sprains were identified in high schoolers (29.6%) than pre-high schoolers (22.8%, *p* = 0.036). Males suffered more fractures than females (35.1% vs. 24.3%, *p* = 0.001). Females sustained more joint sprains (29.1% vs. 21.0%, *p* = 0.012) and contusions/hematomas (12.7% vs. 7.5%, *p* = 0.020). Ankle injuries were more common in females (24.0%) than males (12.0%, *p* = 0.001), while wrist injuries were more prevalent in males (11.7% vs. 7.2%, *p* = 0.034). The most common injury mechanism was apparatus-related, with no differences based on age or sex. Injury types and injured body parts differed depending on age and sex in pediatric hurdle injuries seen in emergency departments. These findings may be helpful for injury prevention and medical care for pediatric hurdlers.

## 1. Introduction

Track and field is one of the most popular sports among young athletes. According to the National Federation of State High School Associations (NFHS), track and field has over one million high school participants, with a near-even split between males and females [1]. Track and field has quickly become one of the fastest-growing sports among middle schoolers as well [2]. Among the many track and field events, hurdling is unique because of the combination of running and jumping over an object at a designated distance. To adjust for different skill and experience levels, the hurdle heights, inter-hurdle distance, number of hurdles, and overall distances are adjusted based on age and sex [3]. Passing over these barriers poses risks, as it forces sudden changes in movement that have potential for causing falls and injuries.

A few studies have been performed to evaluate track and field injuries in young athletes [4,5,6,7,8]. An emergency department-based study conducted on all youth track and field injuries from the 1990s to the early 2000s compared all track and field injuries, which included hurdling [4]. Other studies have looked at the number and proportions of injuries in high school athletes in different track and field events [5] and over several different age groups [6]. In a study by Pierpoint et al., injury types, body parts, injury setting, and injury mechanisms were analyzed over all track and field events but were not categorized according to specific track and field events. A more recent study, which evaluated injured body parts, injury types, and injury mechanisms, may be the first focused on the descriptive epidemiology of pediatric hurdle injuries [8]. However, there is limited literature focused on comparing injured body parts, injury types, and injury mechanisms between youth hurdlers of different sexes and ages, despite the fairly high prevalence of hurdle-related injuries, as compared to other track and field events [4]. Furthermore, hurdle races are divided based upon age and sex, so an understanding of how injuries differ between the different groups is warranted [3]. On the collegiate level, there has been an analysis of sex differences in track and field injuries [9], but hurdling was not included in the study. Sex-specific studies of all track and field injuries, which included those sustained in hurdling, have been performed and have evaluated injured body parts and mechanisms [10,11], but, similarly, these have been limited to collegiate athletes and have not compared injuries across the two sexes. 

Sex and age comparisons have been reported in other studies on sports and injury propensities [12,13,14,15]. For example, it is well-known that females have a higher risk of anterior cruciate ligament (ACL) injuries than males [12,13]. Moreover, according to an age comparison study in football, adolescent athletes are more likely to sustain lower-extremity injuries than their younger counterparts [16]. This information helps medical providers properly prepare for sporting events and may also play a valuable role in developing injury prevention strategies tailored to specific ages and sexes in individual sports [12,13,17]. In summary, even though there are a few studies that have identified both age- and sex-related differences in musculoskeletal injuries [18,19,20], there remains no literature to compare potential age- and sex-related differences in hurdle injuries in the pediatric-aged athletic population [21,22].

In short, combining the growing popularity of track and field with the relatively high injury prevalence in hurdling events, an investigation of hurdle-related injuries based upon age and sex is warranted. The findings of this study may be beneficial to clinicians, coaches, and athletes. Therefore, the purpose of this study was to compare hurdle-related injury types, injured body parts, and injury mechanisms by age and sex among youth. We hypothesized that there would be both age- and sex-based differences in injury types, injured body parts, and injury mechanisms.

## 2. Materials and Methods

### 2.1. Study Design and Data Source

The current study employed a retrospective case review design. Approval of this investigation was received from the institutional review board of the host study institution prior to the commencement of the investigation. The data for this study were retrospectively reviewed and extracted from the National Electronic Injury Surveillance System (NEISS) database. The NEISS of the United States Consumer Product Safety Commission (CPSC) provides data on consumer product-related injuries and sports-and-recreational activity-related injuries that are treated in approximately 100 emergency departments (EDs) across the United States (USA), including eight children’s hospital EDs. These data provide a stratified probability sample of the over 5000 US hospitals with EDs. The data collection starts when a patient is admitted to the ED. An ED staff member enters information into the patient’s medical chart, including age, sex, injury type (diagnosis), affected body part(s), and a description of the event. A NEISS coder reviews all ED records and selects those that meet the criteria for the NEISS. The data are subsequently checked for appropriateness, and once they are approved, the information is added to the NEISS database [23].

In the NEISS database, records of hurdle-related injuries were obtained during a 10-year timeframe (1 January 2008–31 December 2017). Track and field NEISS code 5030 was used to search for hurdle injuries. The eligibility of the data was set by using the following inclusion and exclusion criteria. The two inclusion criteria were: (1) injuries associated with hurdling events and (2) the age of hurdlers was 18 years and under (≤18 years). Conversely, the one exclusion criterium was: (1) erroneous or invalid data records. The injury types and injured body parts were organized according to Timpka et al.’s consensus statement on injury and illness definitions and data collection procedures specialized in track and field studies [24].

### 2.2. Data Classifications and Main Outcome Measures

For injury types, the injury diagnoses were examined. When the injury type was not clearly recorded, two board-certified sports medicine physicians discussed and selected the most reasonable type. When there were several injury types and injured body parts in one hurdler, the data were classified as multiple injury types and multiple injured body parts. Injury mechanisms were determined based on the injury description as principally involving the apparatus (hurdle), the ground, other equipment, or unknown.

The top three injury types, the top three most frequently injured body parts, and the top injury mechanism were treated as the main outcome variables in this study. The three main outcome variables were compared by age (pre-high school vs. high school) and sex (male vs. female). Hurdlers who were 14 years and younger (≤14 years) were classified as pre-high schoolers, while those with an age of 15 years (≥15 years) or older were classified as high schoolers.

### 2.3. Statistical Analysis

The main outcome variables were treated using the frequency function of descriptive analysis. Proportional differences of each of the top three injury types and injured body parts, as well as the top injury mechanism, were analyzed using chi-square (*x*^2^) tests based on age and sex, respectively. The a priori statistical significance was set as *p* < 0.05. Odds ratios (OR) with a 95% confidence interval (95% CI) were also calculated. Statistical Package for Social Science (SPSS, Version 26, SPSS Inc, Chicago, IL, USA) was employed for the current analyses.

## 3. Results

### 3.1. Demographics and Injury Characteristics

A total of 749 records were reviewed. The mean ages of the pre-high school and high school hurdlers were 12.6 ± 1.6 and 16.2 ± 0.9 years, respectively. There were 333 males (44.5%) and 416 females (55.5%). The three most common injury types were traumatic fractures (*n* = 218, 29.1%), joint sprains (*n* = 191, 25.5%), and contusions/hematomas/bruises (*n* = 78, 10.4%). Injury types by age group are expressed in Table 1.

Injury types by sex were previously reported [8]. The top three most-injured body parts were ankles (*n* = 140, 18.7%), knees (*n* = 120, 16.0%), and wrists (*n* = 69, 9.2%). Figure 1 and Figure 2 illustrate the most commonly injured body parts in pre-high schoolers and high schoolers. Injured body parts by age group are included in Table 2.

Descriptive analyses of injured body parts by sex were previously documented [8]. Among four mechanisms, the most common hurdle injury mechanism was apparatus-related, causing a trip or fall or affecting the landing (*n* = 594, 79.0%). The second and third most common mechanisms were the ground (*n* = 47, 6.3%) and other equipment (*n* = 7, 0.9%). A substantial proportion of the mechanisms were unknown in the documentation provided (*n* = 103, 13.8%). 

### 3.2. Injury Types and Injured Body Parts by Age

In comparisons of injury types by age, more traumatic fractures were found in pre-high schoolers (34.1%) than high schoolers (21.5%) [*p* = 0.001, OR: 1.58, 95% CI: (1.23, 2.03), Table 3]. On the other hand, there were more joint sprains among high schoolers (29.6%) than pre-high schoolers (22.8%) [*p* = 0.036, OR: 1.10, 95% CI: (1.01, 1.20), Table 3].

Regarding the injured body parts, no significant differences were detected in a comparison of the top three injured body parts by age, although the proportion of knee injuries appeared greater in high schoolers than in pre-high schoolers (Table 3); however, it was not statistically significant [*p* = 0.055, OR: 1.47, 95% CI: (0.99, 2.17), Table 3].

### 3.3. Injury Types and Injured Body Parts by Sex

In the analyses of injury type by sex, males had more traumatic fractures (35.1%) than females (24.3%) [*p* = 0.001, OR: 1.45, 95% CI: (1.16, 1.81), Table 4]. Conversely, females sustained more joint sprains (29.1%) and contusions/hematomas/bruises (12.7%) than males (joint sprain: 21.0%, contusion/hematoma/bruise: 7.5%) [joint sprain: *p* = 0.012, OR: 1.11, 95% CI: (1.03, 1.21), contusion/hematoma/bruise: *p* = 0.020, OR: 1.06, 95% CI: (1.01, 1.11), Table 4].

Ankle injuries were more common in females (24.0%) than males (12.0%) [*p* = 0.001, OR:2.32, 95% CI: (1.55, 3.46), Table 4]. Conversely, wrist injuries were more prevalent in males (11.7%) than females (7.2%) [*p* = 0.034, OR: 1.62, 95% CI: (1.04, 2.65), Table 4].

### 3.4. Injury Mechanism by Age and Sex

The top injury mechanism, apparatus-related episodes such as tripping, falling, and landing, was compared by age. Pre-high schoolers more frequently sustained their injuries due to an apparatus-related mechanism (81.2%) than high schoolers (75.8%); however, it was not under the a priori statistical significance level [*p* = 0.074, OR: 1.07, 95% CI: (0.99, 1.16)]. In the comparison of this injury mechanism by sex, there was no difference between males (78.7%) and females (79.3%) [*p* = 0.829, OR: 1.03, 95% CI: (0.79, 1.38)] in the hurdle apparatus contributing to the injury mechanism.

## 4. Discussion

The purpose of this study was to compare hurdle-related injury types, injured body parts, and injury mechanisms by age and sex among youth. Our hypothesis was that there would be both age- and sex-based differences in injury types, injured body parts, and injury mechanisms.

After several comparisons, we identified differences according to age and sex in the injury types and injured body parts of pediatric hurdlers, but not in the injury mechanisms. Thus, our hypothesis was generally supported.

### 4.1. Injury Types

Females showed 1.1 times higher odds of joint sprains than males (Table 4). Sex differences in injury rates continue to be a focus in the injury literature [25,26,27]. One of the most-studied ligamentous injuries is the ACL. Females demonstrate higher incidence of ACL injuries than males [28,29]. In order to explain the higher ACL injury incidence in females, sex differences have been discussed as contributing factors in kinetics, kinematics, and neuromuscular control parameters [25]. Additionally, sex differences are also reported in joint hypermobility [30,31], with adolescent female athletes tending to have greater joint hypermobility than males [32]. In addition to movement differences, their joint hypermobility may explain the higher rates of joint sprains in females compared to males among pediatric hurdlers (Table 4). Furthermore, it is hypothesized that athletes who are skeletally immature are more prone to bony injuries due to the properties of bone and higher prevalence of physeal injuries [33,34,35]. Since females undergo skeletal maturation sooner than their male counterparts [33], this could also explain prevalence of soft tissue injuries over bony injuries in the female population. In addition, the males demonstrated approximately 1.5 times greater odds of traumatic fractures relative to females (Table 4). Our findings were in alignment with previous studies, which generally showed higher numbers of fractures in males than females [34,35,36].

Synthesizing these findings, it may be hypothesized that the tighter ligaments found in males [32] could cause there to be less “give” in the ligaments during an acute injury, causing more stress forces to be transmitted to the bones, which, in turn, lead to more fractures and fewer sprains in males than females. For females, greater joint mobility may allow more “give” as stress forces are transmitted across the joint during an injury. Another potential factor in the higher numbers of bony injuries in males is the later onset of puberty in males in comparison to females [33,34,35,37], which corresponds to the fact that males reach skeletal maturity later than females [38,39]. Therefore, as injury forces transmit through the bones, the higher likelihood of having an injury-susceptible physis in males, as opposed to females, may lead to more fractures among males. To support this concept, our age comparison identified about 1.6 times greater odds of fractures in pre-high schoolers than in high schoolers, suggesting that the younger age group is more likely to suffer traumatic fractures than the older group (Table 3). The effects of skeletal maturity on pediatric sports injuries have been previously reported in several studies [18,40,41] and may play a role in the pattern we saw. However, we need more studies to substantiate these hypotheses.

### 4.2. Injured Body Parts

Females demonstrated 2.3 times higher odds than males of having ankle injuries, while males had about 1.6 times greater odds of having wrist injuries than females (Table 4). Our finding is different from the previous hurdle study [42]; however, that study examined professional hurdlers. Our study population, at 18 years of age and younger, is very different from professional-level hurdlers, which likely led to these inconsistent results. Specifically, one possible explanation may be related to the fact that hurdle heights are adjusted based on age and sex [3]. Excluding the youngest age group of 11–12 years, all other age groups have greater hurdle heights for males than females [3]. When males come into contact with the hurdle, it is taller, which requires them to jump higher. A higher jump can also mean a longer distance to fall to the ground. This could potentially lead to the upper extremities, including wrists, being used to attenuate the impact force from the ground during the fall, which subsequently leads to a higher number of wrist injuries in males. Meanwhile, it may be theorized that lower hurdle heights for females may result in less use of their wrists and outstretched arms but relatively more ankle injuries. To verify this notion, more studies are warranted.

Moreover, according to a recent study, there is an association between lower-extremity weaknesses, including hip strength and acute ankle injury risk, especially in females [43]. Another investigation reported that ankle injuries can be reduced with prophylactic exercises and bracing [44]. Synthesizing this evidence in conjunction with our study results suggests a role for sex-specific injury prevention strategies for pediatric hurdlers. For instance, for females, lower-extremity strengthening and ankle supports such as bracing and/or taping may be beneficial. To avoid wrist injuries in males, injury prevention strategies including wrist bracing, proprioception training, and even consideration of altering hurdle height or hurdle rigidity may be applicable.

Even though hurdling is considered a non-contact activity that only involves the lower extremity, there was a notable number of head injuries (4.7%) that occurred. This is surprising, given the non-contact nature of the event, but it suggests that the head is at risk for injury during hurdling events. The proportion of head injuries was higher than other body parts that are in constant motion during hurdling, such as the lower back (2.0%), hips/pelvis (2.7%), and thighs (2.5%). The proportion of head injuries was also equivalent to that of the lower leg (4.8%) and foot/toe (4.8%), both of which undergo regular impaction in hurdling. The study on pediatric track and field injuries performed by Reid et al. also found a greater number of head injuries than expected in hurdling when compared to other track and field events [4]. Among the head injuries found in our study, just over half were diagnosed as concussions. Previous studies have not specifically studied concussions in hurdling athletes, but they have mixed results on the risk of concussions in track and field. A recent report by Tsushima et al. comparing concussion risks for youth athletes suggests that track and field ranks the fourth highest, behind wrestling, cheerleading, and football, for the risk of concussions [45]. However, this study did not differentiate between the various events of track and field, an endeavor which should be undertaken due to the wide variety of the events. In contrast, another study that focused on middle school athletes only found that track and field had the lowest incidence of concussions [46]. A study evaluating the risk of concussions between the different track and field events is warranted. Even as track and field events are distinct, so are positions in team sports. Similar studies have been performed in team sports to examine concussion risks based upon a player’s position [45]. This information would be valuable for medical personnel involved in youth track and field. Further studies to investigate methods to reduce the number of head injuries in pediatric hurdle events should also be pursued.

### 4.3. Injury Mechanism

Although our hypothesis was non-directional, we expected higher proportions of the pre-high schoolers (81.2%) than the high schoolers (75.8%) to sustain injuries due to the apparatus inducing a trip or fall, but the numbers were above the a priori statistically significant level (*p* = 0.074). Our rationale was based on the generally limited hurdling experiences of pre-high schoolers relative to high schoolers. Additionally, the majority of the pre-high schoolers were undergoing physical maturation, including height increases [37], which were linked to many sports injury cases in pediatric athletes [47,48]. As previously mentioned, injury prevention methods could focus on lowering hurdle heights and/or including fewer total hurdles in the race for pediatric hurdlers. Additionally, the hurdle industry may need to manufacture a new hurdle design to ensure the safety of pediatric hurdlers. Such an apparatus could potentially prevent pediatric hurdlers from experiencing various serious injuries while maintaining the core components of the hurdle competitions.

### 4.4. Limitations

Our study has several limitations. First, our data were from the NEISS database, which included only injuries that were presented in a designated NEISS ED. Although the NEISS was designed to obtain a national sample, it only includes about 100 hospitals of the more than 5000 national hospitals. Second, the NEISS database sometimes had incomplete, inconclusive, or insufficient records. To address this type of issue, multiple sports medicine providers made every effort to provide clinically accurate judgements in order to interpret the data used in this study. In cases where there were obvious omissions, the providers worked towards a consensus with professional expertise. When discrepancies still existed, those data were removed from the study. Finally, the NEISS data only include injuries that were presented at the ED. Many musculoskeletal injuries might have been initially evaluated by athletic trainers, coaches, and/or parents without warranting an evaluation in the ED. Therefore, our results are not generalizable to all pediatric hurdle-related injuries, especially on-site treatable, non-emergency musculoskeletal and overuse injuries.

## 5. Conclusions

Among pediatric hurdle injuries recorded in the emergency room, both the injury types and the injured body parts differed by age and sex. There were no sex- or age-based differences in the primary injury mechanism, as hurdle apparatus-related trips, falls, and landings were the main sources of injuries. These findings may be helpful for injury prevention, medical care, and event planning focused on pediatric hurdlers. Future studies that include on-site injury surveillance are warranted in order to capture non-ED hurdle related injuries.

## Figures and Tables

**Figure 1 sports-11-00065-f001:**
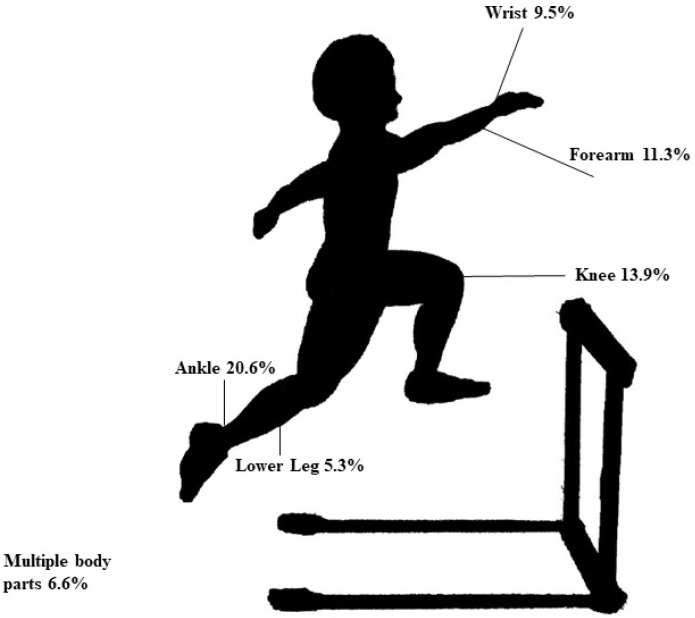
Relative (%) numbers of injured body parts in pre-high school hurdlers.

**Figure 2 sports-11-00065-f002:**
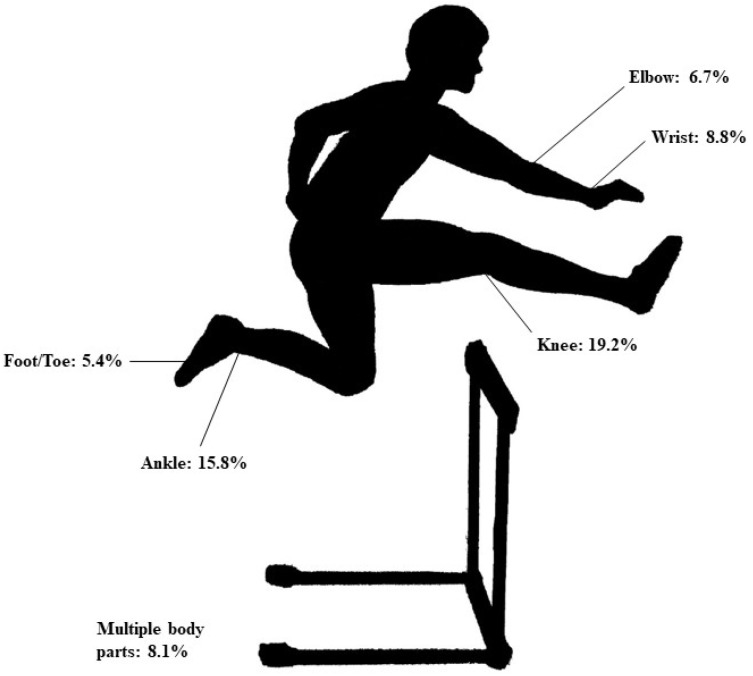
Relative (%) numbers of injured body parts in high school hurdlers.

**Table 1 sports-11-00065-t001:** Absolute and relative (%) numbers of injury types in pre-high school and high school hurdlers.

	All Hurdlers	Pre-High School	High School
**Bony Injury**	**221 (29.5%)**	**154 (34.1%)**	**67 (22.6%)**
Fracture (Traumatic)	218 (29.1%)	154 (34.1%)	64 (21.5%)
Growth Plate Disturbance	2 (0.3%)	0 (0.0%)	2 (0.6%)
Other Bone Injuries	1 (0.1%)	0 (0.0%)	1 (0.3%)
**Joint Injury**	**205 (27.4%)**	**110 (24.3%)**	**95 (32.0%)**
Arthritis/Synovitis/Bursitis	2 (0.3%)	1 (0.2%)	1 (0.3%)
Joint Sprain	191 (25.5%)	103 (22.8%)	88 (29.6%)
Meniscus Lesion/Cartilage	1 (0.1%)	1 (0.2%)	0 (0.0%)
Subluxation/Dislocation	11 (1.5%)	5 (1.1%)	6 (2.1%)
**Muscle/Tendon Injury**	**58 (7.7%)**	**30 (6.7%)**	**28 (9.4%)**
Muscle Cramps/Spasms	1 (0.1%)	0 (0.0%)	1 (0.3%)
Strain/Muscle Tear/Muscle Rupture	53 (7.1%)	27 (6.0%)	26 (8.8%)
Tendinosis/Tendinopathy	4 (0.5%)	3 (0.7%)	1 (0.3%)
**Soft Tissue**	**124 (16.6%)**	**74 (16.4%)**	**50 (16.8%)**
Contusion/Hematoma/Bruise	78 (10.4%)	51 (11.3%)	27 (9.1%)
Fasciitis/Aponeurosis	1 (0.1%)	1 (0.2%)	0 (0.0%)
Laceration/Abrasion/Other Skin Lesion-Related Damages	45 (6.0%)	22 (4.9%)	23 (7.7%)
**Others**	**141 (18.8%)**	**84 (18.6%)**	**57 (19.2%)**
Concussion	18 (2.4%)	12 (2.7%)	6 (2.1%)
Dental Injury/Broken Tooth	1 (0.1%)	1 (0.2%)	0 (0.0%)
Miscellaneous	72 (9.6%)	47 (10.4%)	25 (8.4%)
Multiple Diagnoses	50 (6.7%)	24 (5.3%)	26 (8.8%)
**Total:**	**749 (100%)**	**452 (100%)**	**297 (100%)**

Anatomic categories are bolded and comprise the of injury types immediately below them.

**Table 2 sports-11-00065-t002:** Absolute and relative (%) numbers of injured body parts in pre-high school and high school hurdlers.

	All Hurdlers	Pre-High School	High School
**Head/Trunk**	**83 (11.1%)**	**46 (10.2%)**	**37 (12.4%)**
Head	35 (4.7%)	22 (4.9%)	13 (4.4%)
Face	15 (2.0%)	7 (1.6%)	8 (2.7%)
Neck/Cervical Spine	4 (0.5%)	1 (0.2%)	3 (1.0%)
Upper Back/Thoracic Spine	1 (0.1%)	0 (0.0%)	1 (0.3%)
Sternum/Ribs	4 (0.5%)	4 (0.9%)	0 (0.0%)
Abdomen	4 (0.5%)	1 (0.2%)	3 (1.0%)
Lower Back/Lumbar Spine	15 (2.0%)	10 (2.2%)	5 (1.7%)
Sacrum	5 (0.7%)	1 (0.2%)	4 (1.4%)
**Upper Extremity**	**232 (31.0%)**	**149 (33.0%)**	**83 (28.0%)**
Shoulder/Clavicle	31 (4.1%)	16 (3.5%)	15 (5.1%)
Upper arm	5 (0.7%)	4 (0.9%)	1 (0.3%)
Elbow	42 (5.6%)	22 (4.9%)	20 (6.7%)
Forearm	66 (8.8%)	51 (11.3%)	15 (5.1%)
Wrist	69 (9.2%)	43 (9.5%)	26 (8.8%)
Hand	13 (1.7%)	8 (1.8%)	5 (1.7%)
Finger	6 (0.8%)	5 (1.1%)	1 (0.3%)
**Lower Extremity**	**376 (50.2%)**	**223 (49.3%)**	**153 (51.5%)**
Hip/Pelvis	20 (2.7%)	8 (1.8%)	12 (4.0%)
Groin	5 (0.7%)	4 (0.9%)	1 (0.3%)
Thigh	19 (2.5%)	11 (2.4%)	8 (2.7%)
Knee	120 (16.0%)	63 (13.9%)	57 (19.2%)
Lower Leg	36 (4.8%)	24 (5.3%)	12 (4.0%)
Ankle	140 (18.7%)	93 (20.6%)	47 (15.8%)
Foot/Toe	36 (4.8%)	20 (4.4%)	16 (5.4%)
**Others**	**58 (7.7%)**	**34 (7.5%)**	**24 (8.1%)**
Multiple Body Parts	54 (7.2%)	30 (6.6%)	24 (8.1%)
Missing Data	4 (0.5%)	4 (0.9%)	0 (0.0%)
**Total**	**749 (100%)**	**452 (100%)**	**297 (100%)**

Anatomic regions are bolded with specific injury locations immediately below them.

**Table 3 sports-11-00065-t003:** Comparison of Injury Types and Injured Body Parts by age.

	Pre-High Schoolers	High Schoolers	*p*-Value	OR with 95%CI
**Injury Types**				
Fractures (Traumatic)	34.1%	21.5%	0.001 *	1.58 (1.23, 2.03)
Joint Sprain	22.8%	29.6%	0.036 *	1.10 (1.01, 1.20)
Contusion/Hematoma/Bruise	11.3%	9.1%	0.337	1.24 (0.80, 1.93)
**Injured Body Parts**				
Ankle	20.6%	15.8%	0.103	1.30 (0.94, 1.79)
Knee	13.9%	19.2%	0.055	1.07 (0.99, 1.14)
Wrist	9.5%	8.8%	0.725	1.09 (0.68, 1.73)

% indicates the presence of injured body parts and injury diagnoses. * is a statistical significance. The OR with 95% CI was calculated using lower rates (%) as a baseline. Category headings are bolded.

**Table 4 sports-11-00065-t004:** Comparison of Injury Types and Injured Body Parts by sex.

	Males	Females	*p*-Value	OR with 95%CI
**Injury Types**				
Fracture (Traumatic)	35.1%	24.3%	0.001 *	1.45 (1.16, 1.81)
Joint Sprain	21.0%	29.1%	0.012 *	1.11 (1.03, 1.21)
Contusion/Hematoma/Bruise	7.5%	12.7%	0.020 *	1.06 (1.01, 1.11)
**Injured Body Parts**				
Ankle	12.0%	24.0%	0.001 *	2.32 (1.55, 3.46)
Knee	14.4%	17.3%	0.283	1.04 (0.97, 1.10)
Wrist	11.7%	7.3%	0.034 *	1.62 (1.03, 2.56)

% indicates the presence of injured body parts and injury types. * is a statistical significance. The OR with 95% CI was calculated using lower rates (%) as a baseline. Category headings are bolded.

## Data Availability

Data for this study are publicly available at https://www.cpsc.gov/Research--Statistics/NEISS-Injury-Data (accessed on 1 May 2018).

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
