# Peer review of "Age and Sex Comparisons in Pediatric Track and Field Hurdle Injuries Seen in Emergency Departments of the US"

_sports, 2023, doi:10.3390/sports11030065_

Round 1

Reviewer 1 Report

In summary, this is a study investigating epidemiology of pediatric hurdle injuries. While the epidemiology of overall track and field injuries has been somewhat well studied previously, information on each discipline has been relatively lacking. Manuscript is well written and results are well presented based on the available data. Please see below for minor comments. 

Title: I recommend adding "track and field hurdle injuries seen in the emergency department" since the data derived from the ED, and it may not fully capture the whole spectrum of hurdle injuries. 

Abstract: please include quantitative results

Introduction/Methods/Results/Discussion are well written. Limitations are well recognized.

One thing I would suggest adding is while we see that injuries occur more frequently in lower extremity than in upper extremity, there seems to be good number of head injuries (4.7%) in this population, which is as high as foot/toe and lower leg in terms of proportion. There was a recent NYT article on head injury in olympian hurdler (https://www.nytimes.com/2022/12/17/sports/running-hurdling-head-injuries.html). I wonder if the authors can further describe what these head injuries were. Also, I recommend that authors attempt to acknowledge while head injuries are rare in track and field athletes (or hurdlers), clinicians should not undermine or underestimate them since concussion is not often thought of in track and field athletes. 

Reviewer 2 Report

1.       This study is about hurdle injuries in the USA.It should be reflected in the  title and objective

2.       The population in this study was younger athletes below age 18. Can they be called pediatric athletes

3.       Is this the first study on hurling injuries in this population?

4.       The introduction has to be improved. It should add studies on hurdle injuries in various other populations.

5.       Exclusion criteria 1  is the replication of inclusion criteria

6.       Result and discussion is satisfactory 

Round 2

Reviewer 2 Report

The authors replied to all the comments satisfactorily. The paper has been improved substantially. No further comments.

It can be accepted for publication.